# PrivShap: A Finer-granularity Network Linearization Method for Private Inference

**Xiangrui Xu**                                                          *xxu002@odu.edu*
*Department of Computer Science*
*Old Dominion University*

**Zhenzhen Wang**                                                    *zwang218@jhu.edu*
*Department of Biomedical Engineering*
*Johns Hopkins University*

**Rui Ning**                                                              *rning@odu.edu*
*Department of Computer Science*
*Old Dominion University*

**Chunsheng Xin**                                                    *cxin@iastate.edu*
*Department of Computer Science*
*Iowa State University*

**Hongyi Wu**                                                        *mhwu@arizona.edu*
*Department of Electrical and Computer Engineering*
*University of Arizona*

**Reviewed on OpenReview:** *https://openreview.net/forum?id=7TliYmJr2m*

## Abstract

Private inference applies cryptographic techniques like homomorphic encryption, garble circuit and secret sharing to keep both sides privacy in a client-server setting during inference. It is often hindered by the high communication overheads, especially at non-linear activation layers such as ReLU. Hence ReLU pruning has been widely recognized as an efficient way to accelerate private inference. Existing approaches to ReLU pruning typically rely on coarse hypothesis, which assume an inverse correlation between the importance of ReLU and linear layers or shallow activation layers have less importance for universal models, to assign the budgets according to the layer while preserving the inference accuracy. However, these assumptions are based on limited empirical evidence and can fail to generalize to diverse model architectures. In this work, we introduce a finer-granularity ReLU budget assignment approach by assessing the layer-wise importance of ReLU with the Shapley value. To address the computational burden of exact Shapley value calculation, we propose a tree-trimming algorithm for fast estimation. We provide both theoretical guarantees and empirical validation of our method. Our extensive experiments show that we achieve better efficiency and accuracy than the state-of-the-art across diverse model architectures, activation functions, and datasets. Specifically, we only need $\sim 2.5 \times$ fewer ReLU operations to achieve a similar inference accuracy and gains up to $\sim 8.13\%$ increase on inference accuracy with similar ReLU budgets.

## 1 Introduction

The growth of AI-driven client-server technologies Xing & Zhao (2024a;b;c); Wang et al. (2024); Zhu et al. (2025); Xu et al. (2024a; 2025); Bai et al. (2025), has significantly advanced Machine Learning as a Service (MLaaS) Xu et al. (2024c;b); Mishra et al. (2020); Rathee et al. (2020) in academia and industry. However,

privacy concerns have led to the emergence of cryptographic techniques for private inference (PI), where the server processes encrypted client input while retaining the machine learning model, ensuring data privacy throughout the inference process.

Existing private inference frameworks often involve cryptographic techniques such as Homomorphic Encryption (HE) Fan & Vercauteren (2012); Cheon et al. (2017), Garbled Circuits (GC) Bellare et al. (2012), and Secret Sharing (SS) to maintain the privacy of both the client and the server. For example, CryptFlow2 Rathee et al. (2020) uses HE for convolutional computations and SS for non-linearity computations, while Gazelle Juvekar et al. (2018) employs GC for ReLU operations. Compared to linear computations, the use of GC and SS in non-linear functions like ReLU requires significantly higher communication costs and bandwidth. This increased overhead leads to longer communication latency during client-server interactions in privacy-preserving inference, making efficiency a significant concern.

This challenge has prompted a range of efforts to develop more efficient privacy-preserving models by reducing the number of ReLU operations, thereby decreasing communication and latency overhead in secure client-server interactions. A series of network linearization works that utilizing various methods are summaried in Table.1. Notably, CryptoNAS Ghodsi et al. (2020) and Sphynx Cho et al. (2022a) use the neural architecture search (NAS) technique to find the optimum network architecture. However, such methods usually lead to significant accuracy drops. To rectify such shortcomings, gradient-based and manual methods, such as SNL Cho et al. (2022b) and DeepReDuce Jha et al. (2021), are proposed to prune pixel-wise ReLU operations by adding $l_1$ regularizations during training. However, these methods assume a uniform importance distribution of ReLU operations across different layers, which leads to only a suboptimal balance point between accuracy and communication efficiency. Moreover, it requires extra-long training epochs to reach a low ReLU operation budget. To find a better trade-off between accuracy and efficiency, SENet Kundu et al. (2023) proposes the hypothesis that the importance of ReLU layers is inversely correlated with the corresponding linear layers based on the sensitivity measured by accuracy loss when pruned. Based on this hypothesis, they propose customized layerwise allocation of ReLU budgets based on inferred ReLU importance, which is calculated by one minus the linear layer gradient. Arguably, this hypothesis is only based on limited empirical evidence and not guaranteed to be generalizable to diverse model architectures and settings.

**Our contributions:** We introduce PrivShap, a Shapley Value (SV)-based ReLU pruning method, which provides fine-grained and accurate ReLU importance estimation and assigns ReLU budgets based on this estimation. To the best of our knowledge, this is the first work to estimate the layer-wise importance of ReLU operations using Shapley values Shapley (1953), challenging the ReLU importance hypothesis presented in the SOTA Kundu et al. (2023). To alleviate the computational cost of exact Shapley value calculation with $\mathcal{O}(2^n)$ time complexity, we propose a double-trimming strategy for fast estimation. This method effectively reduces the number of coalitions by trimming the less important subsets of ReLU layer combinations that contribute marginally to the final SV results. Moreover, unlike related works that use $l_1$ regularization to implicitly reduce ReLU operations, PrivShap employs a projected gradient descent pruning strategy to enforce the layer-wise ReLU operations within certain budgets. This approach enables more precise control over the optimization goal during pruning and helps maintain accuracy. We conduct extensive experiments on various models with various activation functions like SiLU/ReLU, including ResNet He et al. (2016) and Wide Residual Networks Zagoruyko (2016), on CIFAR-10, CIFAR100, and Tiny-ImageNet. Experimental results show that our SV-based ReLU budget can yield the best accuracy-efficiency trade-off. We only need up to $\sim 2.5\times$ fewer ReLUs to achieve similar accuracy and gain up to $\sim 8.13\%$ accuracy improvement with a similar ReLU budget compared to prior arts and can be applied to activation functions and models.

## 2 Preliminaries

### 2.1 Cryptographic primitives

We briefly describe the relevant cryptographic primitives in this section.

Table 1: Comparison between existing approaches in yielding efficient models to perform PI. Only our method has fine-grained relu budget allocation for higher model performance and efficiency. PA denotes polynomial approximation on ReLU layer. NAS denotes as network architecture search.

| | Method used | Granularity | Fine-grained? |
|---|---|---|---|
| Sphynx Cho et al. (2022a) | NAS | layer | × |
| CryptoNAS Ghodsi et al. (2020) | NAS | layer | × |
| DELPHI Mishra et al. (2020) | NAS +PA | layer | × |
| SAFENet Lou et al. (2021) | NAS +PA | channel | × |
| DeepReDuce Jha et al. (2021) | Manual | layer | × |
| SNL Cho et al. (2022b) | $l_1$-normalization | pixel, channel | × |
| SENet Kundu et al. (2023) | layer-importance based | channel, pixel | ✓ |
| Ours | Shapley value based | channel, pixel | ✓ |

**Homomorphic Encryption:** HE is a public key encryption scheme that allows computations to be performed on ciphertexts, enabling the results to be decrypted without needing to access the plaintexts. The encryption function $E$ produces a ciphertext $t$ from a plaintext message $m$ using a public key $pk$, expressed as $t = E(m, pk)$. In PI, the results of linear operations can be obtained homomorphically through $m_1 \circ m_2 = D(t_1 \star t_2, sk)$, where $\circ$ represents a linear operation, $\star$ is its corresponding homomorphic operation, and $t_1$ and $t_2$ are the ciphertexts of $m_1$ and $m_2$, respectively.

**Garbled Circuits:** GC facilitate secure two-party computation of a Boolean function $f$ without revealing private inputs. The function is represented as a Boolean circuit $C$. The process begins with the garbler, who generates an encoded circuit $\tilde{C}$ and corresponding input labels using the $Garble(C)$ to send $\tilde{C}$ and the labels to the other party who acts as an evaluator. The evaluator evaluates the garbled circuit with the provided labels via $Eval(\tilde{C})$. Finally, the garbler decrypts the labels to get the plain results to share with the evaluator.

**Additive secret sharing**. Given an original message $m$ at party $P \in \{0, 1\}$, one of the two Additive Secret Shares (ASS) is constructed by uniformly sampling randomness $r$ and setting $\langle m \rangle_P = r$, while the other share is formed as $\langle m \rangle_{1-P} = m - r$. To reconstruct the message, one can simply add two shares $m = \langle m \rangle_P + \langle m \rangle_{1-P}$.

## 2.2 Private Inference

Similar to previous works Mishra et al. (2020); Cho et al. (2022b); Kundu et al. (2023), our method follows the *two-party semi-honest* threat model. Specifically, the client $C$ and the server $S$ follow the protocol but attempt to infer each other's input, namely the client's input data and the server's model parameters, during the inference process. To mitigate various threats, existing cryptographic-based inference frameworks often employ an online-offline topology. In this architecture, the client's data-independent components are precomputed during the offline phase. For linear computations, such as matrix multiplication, HE is commonly utilized in the offline stage, as it allows operations like multiplication on ciphertexts without the need for decryption. To ensure the correctness of decryption, ciphertexts must be refreshed after a limited number of operations, which can be accomplished through techniques such as bootstrapping Chillotti et al. (2016) or re-encryption Rathee et al. (2020). However, the computationally intensive nature of non-linear functions in GC results in high costs for operations like ReLU, even in the online phase. For SS-based customized privacy-preserved activation protocols, the frequent communication via Oblivious Transfer cryptographic primitives imposes prolonged inference times. This highlights a critical challenge in balancing efficiency and security in cryptographic computations.

## 2.3 Shapley Value

Shapley Value is a key concept in cooperative game theory, which assumes a set of players collaborate to achieve a total gain, and measures each player's contribution by averaging its marginal contribution across all possible coalitions. Shapley value has been widely applied to model explanation Sundararajan & Najmi

(2020), data evaluation Ghorbani & Zou (2019), and model compression Ghorbani & Zou (2020) as it has the following desirable properties:

- **Zero/Negative Contribution:** The player with zero (or negative) contribution to the model accuracy can be measured if $\forall S \subseteq N/\{i\} : U(S \cup \{i\}) \leq U(S)$, which means adding the element to the subset of other layers does not change or reduce the model performance metric ($N$ denote the complete set, $S$ denote a coalition of elements, and $U$ denote the utility function). The proof is shown in appendix B.1 in Ghorbani & Zou (2020).

- **Efficiency:** The sum of the Shapley values of all ReLU layers equals the value of the grand coalition, so that all the gain is distributed among the ReLU layers: $\sum_{i \in N} SV(i) = SV(N)$, where $SV(N)$ denotes as the value of grand coalition. The proof is shown in appendix B.1 and B.2 in Ghorbani & Zou (2020).

However, the calculation of exact SV requires exponential computational costs to enumerate all possible coalitions, and existing works use Monte Carlo sampling (MCs) Ghorbani & Zou (2019) or Multi-Armed Bandit (MAB) Ghorbani & Zou (2020) to approximate it. However, such works still need an extra long time to converge via a large number of random samples.

### 2.4 ReLU linearization in Private Inference

Existing approaches to reduce ReLU operations in neural networks for private inference focus on either re-designing models or optimizing architectures. SAFENet Lou et al. (2021) achieves finer control by applying channel-wise substitutions and mixed-precision activations, measuring activation importance via $l_1$ normalization from preceding layers. CryptoNAS Ghodsi et al. (2020) redesigns networks using evolutionary NAS to minimize ReLU operations, while Sphynx Cho et al. (2022a) enhances this with differentiable NAS for more efficient privacy-preserving models. DeepReDuce Jha et al. (2021) takes a manual approach, pruning redundant ReLU layers from standard models. Recent efforts Cho et al. (2022b) also apply $l_1$ regularization to reduce ReLU operations, optimizing the trade-off between accuracy and non-linearity. However, these methods are sensitive to hyperparameters, often leading to suboptimal solutions and failing to meet specific ReLU budget targets. Furthermore, those works overlook the potentially different importances of ReLU operations across layers and only assign a global ReLU budget for the entire model.

SENet Kundu et al. (2023), on the other hand, designs a fine-grained, layerwise ReLU budget allocation strategy. Their method is based on a hypothesis that the importance of each activation layer is inversely correlated with the pruning sensitivity [1] of its corresponding layer with weight and bias parameters. They then assign the ReLU budget by measuring the pruning sensitivity. However, this hypothesis is only based on their empirical observation in a small experiment of training ResNet18 on CIFAR-100. It is not convincing that this is a generalizable rule in other datasets and model variants without further verification. Furthermore, they define the importance of ReLU layers as the accuracy of retaining only one ReLU layer at a time and removing all others, which overlooks the natural interactions between different ReLU layers in a model. Shapley value provides a compelling alternative to measure layerwise ReLU importance rigorously and directly.

## 3 Method

In this section, we introduce our PrivShap method for fine-granularity network linearization method that includes two parts. First, Sec. 3.1 introduces our method uses SV as indicators and tree-trimming method to allocate a ReLU budget across layers. Second, Sec. 3.2 introduces gradient-projection pruning to remove activations at a finer granularity. To better formulate the interactions and cooperation of different ReLU layers, we treat them as cooperative players and assess their averaged marginal contribution to preserving the inference accuracy. More formally, let $N$ denote the set of all ReLU layers, $S$ denote a coalition of

---

[1]Pruning sensitivity denotes the accuracy reduction caused by pruning a certain ratio of parameters from it Ding et al. (2019).

ReLU layers (i.e., a subset of ReLU layers), and $U$ denote the performance assessment function, specifically assessing the model accuracy of preserving only a coalition of ReLU layers and replacing all other ReLU layers as identity function [2]. The Shapley value of the $i$-th ReLU layer is then defined as:

$$SV_i = \frac{1}{|N|} \sum_{S \subseteq N \setminus \{i\}} \frac{U(S \cup i) - U(S)}{\binom{|N|-1}{|S|}} \tag{1}$$

The Shapley value takes into account the interactions between the ReLU layers. As a simple example, suppose that two layers improve performance only if they are both present but harm performance if they are present individually, estimating their importance by preserving each of them at a time (as Kundu et al. (2023) did) might give misleading results. Shapley value, on the other hand, considers all possible coalitions.

### 3.1 Fine-grained ReLU importance Estimation

However, the exact SV measurement for the ReLU layers works for the model with a few layers. The computational complexity grows quickly as the model goes deeper, becoming prohibitively expensive even for ResNet-18. This is mainly due to an exponential number of coalitions $\mathcal{S}$, and computing the marginal contribution for all of them is extremely time-consuming. Existing works have employed methods such as

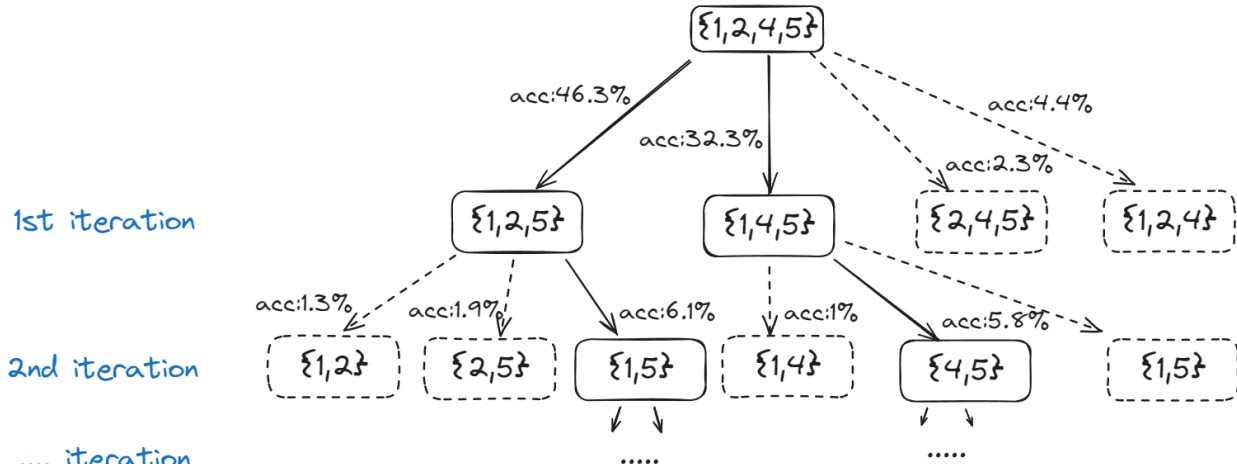

Figure 1: The process of calculating the estimated shapley value of 3rd ReLU layer in a given example 5-ReLU layer model, the dashed branches are trimmed as the remaining accuracy below pre-defined threshold ($< 5\%$). Each iteration will generate one layer of child subsets with removing one element.

MCs and MAB algorithms to approximate the SV of data points or neurons once specific convergence criteria are met. Although MCs and MAB algorithms can provide approximate solutions, they do not eliminate the inherent computational burden associated with effectively estimating SV.

To avoid an exponential subset search space, the MCs and MAB first sample subsets of ReLU layers based on the given model architecture. After the candidate model architecture is determined, the model is trained via hundreds of epochs to converge on the given datasets. Then the layer-wise SVs are updated with the new model accuracy data point. Such a sampling process is repeated until the calculated SVs are converged or within a given small threshold, as shown in Fig. 2 left. Therefore, the sampling trials and heavy training process within each trials are inevitably leading to an unacceptably long training process.

To reduce heavy computational cost, we propose a tree-trimming method to estimate the SV of each ReLU layer, as shown in Fig. 2 right. Our method is based on an observation that all the ReLU layers have positive contributions to the total accuracy and there are few "free-riders" layers. An intuitive explanation for this is that even if ReLU operations at some certain pixels might be harmful to the model accuracy Li et al. (2022),

---

[2]All the layers with weight and biases are preserved

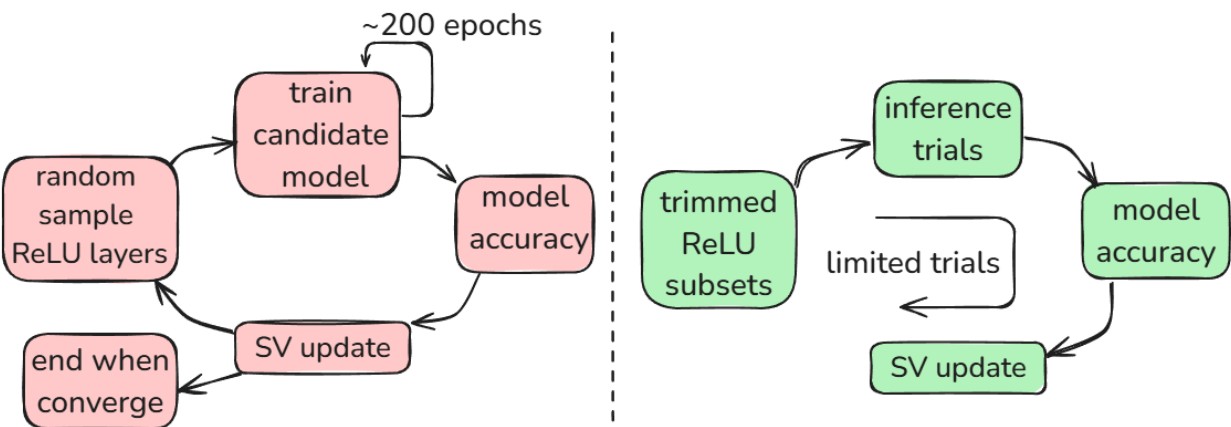

Figure 2: The SV estimation process via Monte-Carlo sampling (left). The process involves unbounded sampling trials and heavy training for each candidate model, while PrivShap (right) only requires inference with limited trials with bounded error (see in Fig. 1 and Sec. 3.1).

an entire layer (i.e., a set of pixel-wise ReLU operations), can be less likely to have a negative contribution. This is also aligned with our empirical observation: the accuracy can plummet to a very low number even if only removing few ReLU layers from a trained model. This also makes such low accuracy model irreversible to high accuracy model by just adding one important activation layer. We start with a model trained in a standard, non-ReLU pruning manner. After the model is trained, we calculate the (approximated) SV for its $i$-th ReLU layer by building a tree, where each node denotes an inference. The root node represents all ReLU layers excluding the $i$-th one (i.e., $N \setminus \{i\}$), and each node represents a coalition of ReLU layers containing one fewer ReLU layer than its father node. At each node with the corresponding coalition $S$, we calculate the marginal contribution of the $i$-th layer to inference accuracy, denoted as $U(S \cup \{i\}) - U(S)$. Importantly, we trim all the child nodes of $S$ if $U(S)$ is below a certain threshold, as the marginal contributions in these coalitions (i.e., child nodes of $S$) are even lower, resulting in a negligible perturbation of the SV. Figure 1 illustrates this trimming process, where the dashed nodes denote those trimmed. Since each ReLU layer makes similar contribution and those layers are minority in mainstream models, the valid nodes are bounded within very few iterations. Thus, we gain time efficiency by eliminating the heavy training and unbounded sampling trials, which makes our method suited for solving network linearization issue. The detailed runtime comparison is shown in Sec. 4.3.

**Preliminary Theoretical Analysis** To demonstrate the accuracy of our approximation, we provide a preliminary theoretical analysis. We aim to show that given a coalition of ReLU layers $S$ where its utility $U(S)$ is below a certain pruning threshold $\alpha$, pruning $S$ and all its subsets will only have a negligible perturbation to the overall SV. We start with a natural observation that $\forall S' \subseteq S$ and $U(S) \leq \alpha$, then the marginal gain satisfies $U(S' \cup i) - U(S') \leq \beta$, where $\beta \ll \alpha$. The perturbation caused by pruning $S$ and all its subsets can be formulated as:

$$
\begin{aligned}
\Delta SV_i &= \frac{1}{|N|} \sum_{S' \subseteq S} \frac{U(S' \cup i) - U(S')}{\binom{|N|-1}{|S'|}} \leq \frac{\beta}{|N|} \sum_{S' \subseteq S} \frac{1}{\binom{|N|-1}{|S'|}} \\
&= \frac{\beta}{|N|} \sum_{0 \leq k \leq |S|} \frac{\binom{|S|}{|S'|}}{\binom{|N|-1}{|S'|}} < \frac{\beta}{|N|} \sum_{0 \leq k \leq |S|} 1 = \beta \frac{(|S|+1)}{|N|}
\end{aligned}
\tag{2}
$$

Even when pruning at the nodes close to higher level of the tree, this perturbation remains negligible, as $\beta$ is typically two orders of magnitude smaller than the pruning threshold $\alpha$.

---

**Algorithm 1** Layer-wise ReLU importance Estimation

---

**Input:** one-layer subsets $\mathcal{N}$, the set of ReLU layers $\mathcal{S}$, model $\mathcal{M}$.
**Output:** ReLU importance (SV) for each relu layer $sv$.

1:   $sv = 0$
2:   $queue \leftarrow init\ with\ \emptyset$
3:   $subsets \leftarrow generating\ all\ subsets\ of\ N$              ▷ in elements descending order
4:   $queue \leftarrow subsets$
5:   **while** $queue\ is\ not\ \emptyset$ **do**
6:      $mask \leftarrow pop\ the\ mask\ combination\ from\ queue$
7:      $U\_s \leftarrow get\ accuracy\ of\ masked\ model$
8:      $all\_mask \leftarrow add\ the\ candidate\ layer$
9:      $U\_si \leftarrow get\ accuracy\ of\ all\_masked\ model$
10:     **if** $U\_si > threshold$ **then**
11:       **for** child in $find\_subsets(mask)$ **do**             ▷ one element less subsets
12:         **if** child not in queue **then**
13:           $queue.push(child)$
14:         **end if**
15:       **end for**
16:     **end if**
17:     $sv = sv + (U\_si - U\_s)/\binom{N-1}{|\mathcal{S}|}$
18: **end while**
19: $sv = sv/|\mathcal{S}|$

---

## 3.2 Gradient-Projection Pruning Strategy

After obtaining the approximated SV for each ReLU layer, we normalize these values and multiply them by the total ReLU budget to get the layer-wise ReLU budget. To avoid conflicts, we ensure that the allocated budget does not exceed the number of pixel-wise ReLU operations that each layer can provide. We fine-tune the pre-trained standard model (i.e., all ReLU persevered) and prune ReLU layers during fine-tuning. Practically, we consider $x_i \in \mathbb{R}^{d \times m \times n}$ to denote the feature map before the $i$-th ReLU layer, where $d, m, n$ denote the number of channel, width and length, respectively. We attach a binary mask $c_i$ with the same shape as $x_i$ to the $i$-th ReLU layer. The output feature map of the masked ReLU layer can be formulated as

$$x_i' = (1 - c_i) * x_i + c_i * \text{ReLU}(x_i). \tag{3}$$

The masks at every layer are initialized as all-ones as we start with a standard model with all ReLU operation preserved.

Previous works, such as SNL, use the $l_1$ norm to increase the sparsity of $c_i$, given as $\mathbb{L} = \min_{W,C} L(f_{W,C}(X), y) + \lambda(\sum_{i=1}^{b} ||c_i||_1)$, where $W$ denotes the weights and $C = \{c^1, c^2, .., c^b\}$, $b$ denotes the total number of ReLU layers for a given model. However, it experience a significant drop in accuracy when targeting a low ReLU budget since the optimization goal has been shifted from improving accuracy to increasing sparsity. This causes model performance degradation. To address this, we propose a gradient-projection strategy, which remove the $l_1$ norm from the objective function and attach a pruning process after every gradient descent step. To be more specific, we calculate the gradient for $c_i$, which yields a ranking of importance for the $d \times m \times n$ pixel-wise ReLU operations on that layer, where a larger gradient magnitude indicates a higher importance of the ReLU operation at the corresponding location. Next, we retain the top-$k$ important ReLU operations, where $k$ is the layer-wise ReLU budget determined by the approximated Shapley value. The mask $c_i$ is updated accordingly, where pruned ReLU operations are set to 0 and others to 1. This strategy ensures that the layerwise ReLU budget is always met during the entire fine-tuning process. Moreover, the objective function is the sum of cross-entropy loss (to fit ground truth labels) and Kullback–Leibler (KL) divergence loss (to approximate the original model), which ensures the optimization is only focused on improving model performance. The overall pipeline is depicted in Fig. 3.

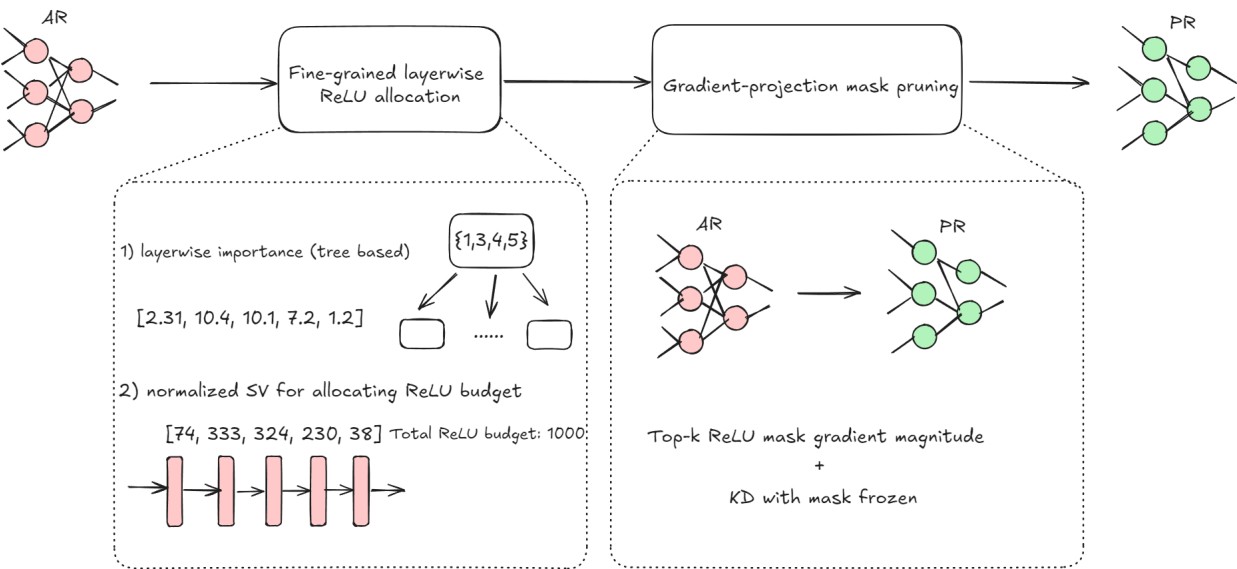

Figure 3: The overview of PrivShap, AR denotes the all-ReLU model and PR denotes the partial-ReLU model. We calculate the approximated SV for each non-linear layer (ReLU) based on the tree search algorithm. The normalized SV of each ReLU layer is used to allocated the layerwise ReLU budget, with given ReLU budget. Then we train the PR model with AR model by keeping top-k ReLU mask gradient magnitude and narrowing the similarity between PR and AR model. Such process applies the generated fine-grained layerwise budget allocation.

## 4 Evaluation

### 4.1 Experiment Setup

To evaluate our method and align with SOTA evaluation setting for clear comparison, we apply PrivShap to models such as ResNet18, ResNet34 with SiLU activation function, and WideResNet-22 in three benchmark datasets: CIFAR-10, CIFAR-100 and Tiny-ImageNet. CIFAR-10 and CIFAR-100 both have images with a resolution of 32×32. CIFAR-10 contains 10 classes with 5,000 training images and 1,000 test images per class, whereas CIFAR-100 includes 100 classes, with 500 training images and 100 test images per class. Tiny-ImageNet has a larger image resolution of 64×64 and consists of 200 classes, each with 500 training and 50 validation images. The experiments are conduct on two servers with an AMD EPYC 7413 24-core Processor 64GB RAM, under WAN (1 GBps, 1 ms latency) and LAN (300MBps, 5 ms latency).

We first train the all-ReLU (AR) models on CIFAR-10, CIFAR-100, and Tiny-ImageNet datasets using the SGD optimizer with an initial learning rate of 0.1, momentum of 0.9, and weight decay of 0.0005. The learning rate decays by a factor of 0.1 at 80 and 120 epochs for CIFAR-10/100, using a batch size of 256. For Tiny-ImageNet, we train for 200 epochs with the same hyperparameters. After this, we apply gradient-projection pruning for partial-ReLU (PR) models with a trimming threshold of 5% and an initial learning rate of 0.001 using the Adam optimizer. Finally, the ReLU masks are frozen and the model parameters are fine-tuned following the distillation process of Kundu et al. (2023).

We follow the latency measurement method of Cho et al. (2022b) and break down the sources of latency into two categories: ciphertext linear operations and GC-based ReLU operations. As the SNL follows the DELPHI framework, the ciphertext linear computation is moved to offline and use BFV scheme for HE with default 4096-slots cryptographic parameter of SEAL library. We set 15 bit-length share for model and 31 bit share for ReLU protocol. For ReLU latency, the wall-clock time for 1000 ReLU operations is measured as t = 0.021 seconds per 1000 ReLUs. We compare our method with multiple baselines including SNL Cho et al. (2022b), DeepReDuce Jha et al. (2021), Sphynx Cho et al. (2022a), SAFENet Lou et al. (2021), CryptoNAS Ghodsi et al. (2020), SENet Kundu et al. (2023) and AutoFHE Ao & Boddeti (2024).

Table 2: The performance of PrivShap and other methods on model architectures with CIFAR-100. 'r' denotes the given ReLU budget, Comm. Saving denotes the ratio of communication costs associated with an AR model to that of the corresponding PR model with reduced ReLUs.

| min$<$r $<$max | model | baseline Acc% | #ReLU(k) | Method | Test Acc% | Comm. Saving |
|---|---|---|---|---|---|---|
| | | | CIFAR-100 | | | |
| 0$<$r$<$100k | VGG11 | 76.84 | 90.8 | Ours | 75.4 | 7.2× |
| | ResNet18 | 78.3 | 75.2 | | 74.9 | 9.4× |
| | ResNet34(SiLU) | 78.23 | 69.6 | | 71.2 | 11.2× |
| | VGG11 | 76.84 | 91.5 | SNL | 70.2 | 5.3× |
| | ResNet18 | 78.3 | 86.3 | DeepReDuce | 68.5 | 10.3× |
| 100k$<$r$<$400k | VGG16 | 78.7 | 110 | Ours | 78.2 | 3.4× |
| | ResNet50 | 78.3 | 160 | | 77.8 | 3.6× |
| | ResNet34(SiLU) | 76.92 | 280 | Sphynx | 71.4 | 2.8× |
| | WRN22-8 | 81.2 | 265 | SAFENet | 78.5 | 4.1× |
| | VGG11 | 76.84 | 150 | CryptoNAS | 70.3 | 2.7× |
| | ResNet34 | 75.15 | 167 | AutoFHE | 72.3 | 4.2× |
| | ResNet18 | 78.3 | 135 | SENet | 74.2 | 4.5× |

Table 3: The performance of PrivShap and other methods on model architectures with CIFAR-10. 'r' denotes the given ReLU budget, Comm. Saving denotes the ratio of communication costs associated with an AR model to that of the corresponding PR model with reduced ReLUs.

| min$<$r $<$max | model | baseline Acc% | #ReLU(k) | Method | Test Acc% | Comm. Saving |
|---|---|---|---|---|---|---|
| | | | CIFAR-10 | | | |
| 0$<$r$<$100k | VGG11 | 93.2 | 80.8 | Ours | 92.5 | 4.5× |
| | ResNet34(SiLU) | 95.8 | 55.2 | | 90.4 | 10.4× |
| | ResNet101 | 95.9 | 49.6 | | 89.2 | 21.7× |
| | VGG11 | 93.2 | 85.5 | SNL | 85.8 | 17.1× |
| | ResNet18 | 95.8 | 75.3 | DeepReDuce | 83.2 | 9.3× |
| 100k$<$r$<$400k | VGG16 | 94.1 | 110 | Ours | 93.2 | 3.1× |
| | ResNet34(SiLU) | 95.8 | 160 | | 95.3 | 3.5× |
| | ResNet18 | 95.9 | 180 | Sphynx | 91.2 | 5.6× |
| | WRN22-8 | 94.6 | 165 | SAFENet | 90.4 | 2.8× |
| | VGG11 | 93.2 | 150 | CryptoNAS | 87.5 | 2.3× |
| | ResNet34 | 94.3 | 168 | AutoFHE | 91.7 | 3.8× |
| | ResNet18 | 95.8 | 125 | SENet | 89.7 | 4.9× |

## 4.2 PrivShap achieves better inference accuracy-efficency trade-off

To demonstrate our method to model performance and inference acceleration on pruned model, we compare our method with gradient-based ReLU pruning (SNL), ReLU budget allocation (SENet), and other SOTA, as shown in Fig. 4. Our method achieves up to an 8.13% accuracy improvement over SENet on ResNet-34 in CIFAR-100 with a ReLU budget of 150k and 5.5% accuracy boost on 7k lower ReLU budget. On Tiny-ImageNet, we observe an improvement in accuracy of 3.1% to 4% in various ReLU budgets. For CIFAR-10, the accuracy gain is around 2%, likely due to the simpler 10-class classification task and high baseline accuracy.

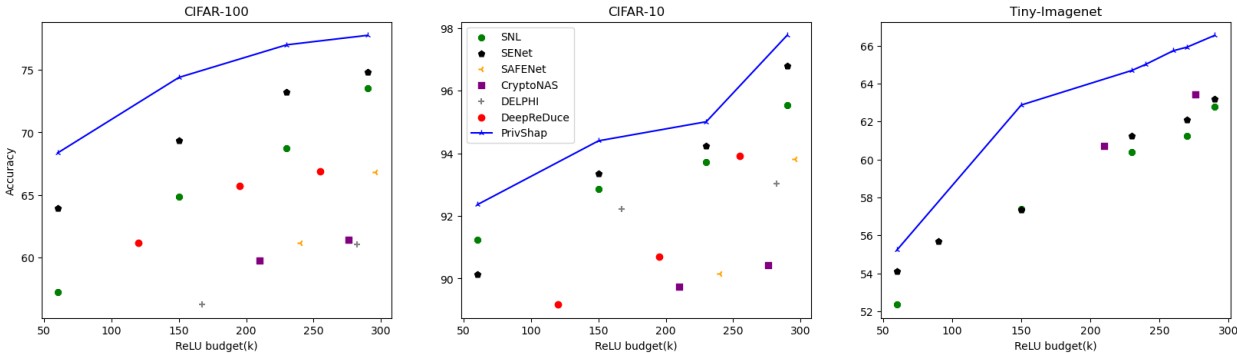

Figure 4: The accuracy performance of PrivShap v.s existing ReLU pruning works on various ReLU-budgets in ResNet-34. Our method achieves better Pareto frontier than existing works on different datasets.

Table 4: The exact and estimated SVs and standard deviation per ReLU layer in ResNet-9 on CIFAR-100 with 5 trials. $\rho$ denotes the Pearson correlation.

| ReLU Layer Index | 1 | 2 | 3 | 4 | 5 | 6 | 7 | 8 | $\rho$ |
|---|---|---|---|---|---|---|---|---|---|
| Exact SV | 10.98 | 11.12 | 11.55 | 11.03 | 10.85 | 10.87 | 6.47 | 0.13 | - |
| Estimated SV(ours) | 8.66 ±.01 | 9.17 ±.02 | 9.24 ±.01 | 9.28 ±.02 | 9.07 ±.02 | 8.45 ±.02 | 5.27 ±.02 | 0.09 ±.02 | **0.997** ±**.02** |
| Estimated SV(MC) | 8.82 ±.05 | 9.02 ±.08 | 8.92 ±.02 | 9.11 ±.03 | 9.32 ±.01 | 8.61 ±.05 | 6.44 ±.02 | 0.11 ±.04 | 0.987 ±.04 |
| Estimated SV(MAB) | 9.41 ±.07 | 10.43 ±.12 | 9.96 ±.08 | 10.17 ±.05 | 9.35 ±.02 | 9.15 ±.06 | 7.33 ±.04 | 0.24 ±.02 | 0.981 ±.06 |

For low ReLU budgets ($\leq 100k$), our method reduces the number of ReLUs by $2.5\times$ and results in a $11.2\times$ communication savings in online latency for ResNet-34 in CIFAR-100, as shown in Table 2, outperforming existing methods in both accuracy and communication efficiency. Our method also demonstrates $1.3\times$ ReLU budgets compared to the SOTA and up to $21.7\times$ communication savings for ResNet-101 in CIFAR-10, as shown in Table 3. Our method also maintains higher accuracy at higher ReLU budgets ($> 100k$) compared to other approaches, achieving around 3.2% more accuracy in ResNet-18 with CIFAR-100, shown in Table 2.

Table 5: The wall-clock time and pruned model accuracy of different SV estimation methods

| Model | Methods | Wall-clock time | Pruned model accuracy | | |
|---|---|---|---|---|---|
| | | | C-100 | C-10 | T-imagenet |
| ResNet-9 | Ours | $\sim$ **5 min** | **76.2%** | **93.8%** | **65.4%** |
| | MC | > 45 min | 71.4% | 89.3% | 57.7% |
| | MAB | > 52 min | 71.9% | 90.6% | 59.6% |
| ResNet-18 | Ours | $\sim$ **35 min** | **78.5%** | **94.3%** | **65.1**% |
| | MC | > 70 min | 73.9% | 91.5% | 59.6% |
| | MAB | > 78 min | 72.4% | 93.1% | 63.7% |
| ResNet-34 | Ours | $\sim$ **45 min** | **77.3%** | **93.5%** | **67.1%** |
| | MC | > 210 min | 72.1% | 90.1% | 58.7% |
| | MAB | > 196 min | 73.6% | 87.9% | 63.2% |

### 4.3 Tree-trimming algorithm outperforms other SV estimation methods

To evaluate the accuracy of our SV estimation using the tree-trimming algorithm, we compare the estimated values with the exact SV. Given the high computational complexity of exact SV calculation, we performed our experiments on a lightweight model, ResNet-9, which consists of eight ReLU layers, on the CIFAR-100 dataset. We use inference accuracy as the utility function and a threshold of 5% in tree-trimming estimation. Table 4 shows the estimated and exact SVs of all ReLU layers, which align with each other, achieving a Pearson correlation of 0.997. We compare with other SV estimation methods, including MC Ghorbani & Zou (2019) and MAB Ghorbani & Zou (2020). They show slightly lower correlations.

Notably, both the exact and estimated SVs show a fluctuating importance trend on ReLU importance, and the last two ReLU layers appear to be the least important compared to the former ones. Previous works, such as SENet, SNL, and DeepReDuce, often assume a consistent trend of layerwise importance. These results are counterexamples, implying that these assumptions might be oversimplified and not able to generalize to different model architectures and downstream tasks.

Moreover, we evaluate the computational efficiency and the accuracy of pruned models using different SV estimation methods across various model architectures and datasets (Table 5). For fair comparison, the MCs and MAB methods calculate the approximated SV for each ReLU pixel in a single run and keep the top-k budget of ReLU pixels based on normalized SV importance. Results show that MC and MAB take $2\times$ to $4.7\times$ more wall-clock time (which denotes the sum of time of calculating the SV for each ReLU layer) than our tree-trimming method. Our method achieves 2.8% to 4.6% higher model accuracy, while only costing about half of the wall-clock time. All these results provide strong evidence that the tree-trimming algorithm we proposed outperforms other SV estimation methods in both accuracy and efficiency.

### 4.4 Trimming threshold sensitivity

As detailed in Sec. 3, our tree-trimming algorithm has a hyperparameter, the threshold for tree-trimming. We now discuss the sensitivity of this threshold. We estimate the SV per ReLu layer with trimming thresholds of 5%, 10%, and 15%, different thresholds on various models and datasets (See Figure 5 - 8). In general, the estimated SVs show consistent trends across different thresholds.

Furthermore, we evaluated how this threshold affects the ultimate inference accuracy. With a 150k ReLU budget, we plot inference accuracy as a function of trimming threshold across different models and datasets in Fig. 9. Not surprisingly, the inference accuracy drops with threshold increases, as a higher threshold leads to inaccurate SV estimation (see details in Sec. 3.1). However, the gradient is reasonably small between 5% and 20%, which implies that the model's performance remains stable if the trimming threshold is chosen from this range.

## 5 Discussion and Conclusion

In this paper, we propose PrivShap, a method for finer-grained network linearization to improve private inference. We show that previous works use over-simplefied layerwise ReLU importance assumptions, such as inverse correlation between ReLU and weight layers, which can not generalize to different models and tasks. We propose to use Shapley Values for fine-grained and accurate ReLU importance estimation and allocate the budget accordingly. As exact Shapley value calculation is extremely computationally expensive, we propose a tree-trimming algorithm for fast Shapley Value approximation, allowing for generalization to larger models with many layers. Additionally, we propose a gradient-projection pruning strategy to enforce the number of ReLU operations per layer below allocated budgets. Our experiments show that PrivShap reduces ReLUs by up to $\sim2.5\times$ with similar accuracy, and improves accuracy by up to $\sim8.13\%$ with comparable ReLU budgets, when compared with the prior arts. Furthermore, when compared with other Shapley value estimation approaches, the proposed tree-trimming method outperforms in both approximation accuracy and efficiency.

Naturally, our method has some limitations. 1) Our work aims to provide a new perception on measuring the importance of activation layers that can guide the pixel-wise activation pruning for fast private inference.

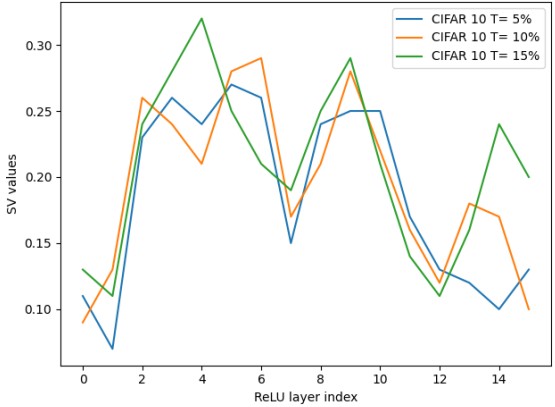

Figure 5: The approximated SV of each ReLU layer in ResNet-18 on CIFAR10.

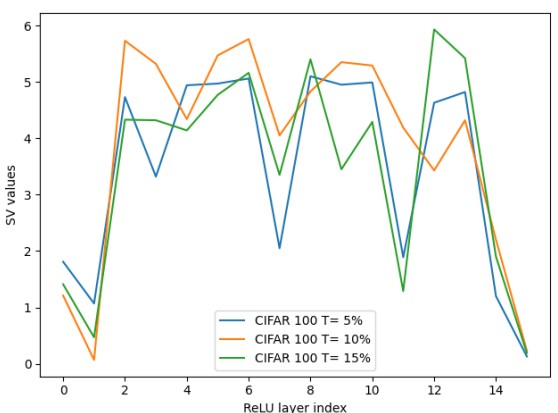

Figure 6: The approximated SV of each ReLU layer in ResNet-18 on CIFAR100.

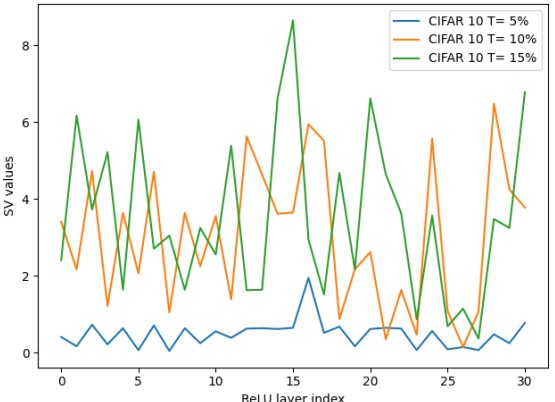

Figure 7: The approximated SV of each ReLU layer in ResNet-34 on CIFAR10.

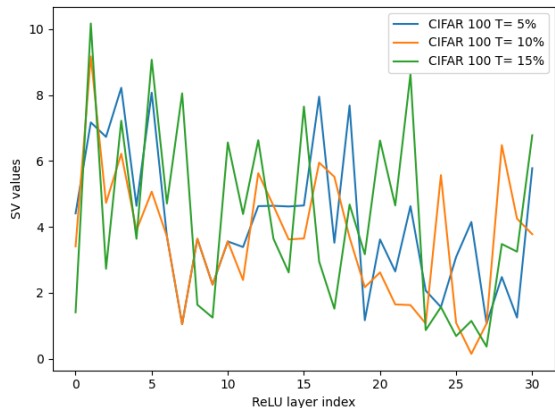

Figure 8: The approximated SV of each ReLU layer in ResNet-34 on CIFAR100.

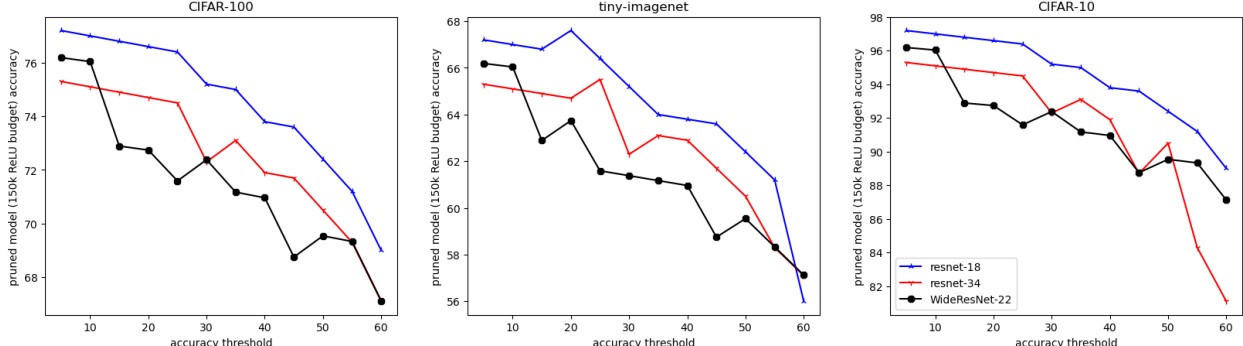

Figure 9: The accuracy performance of PrivShap with different accuracy thresholds in various datasets.

While lack of further experiments on larger model and more complex dataset, our method can be generalized to such scenarios as those models follow similar structure. 2) Our method can be sensitive to the threshold choice to make the optimized trade-off between estimation accuracy and estimation efficiency for specific models. The threshold choice and theoretical analysis remain to be explored in future study. 3) Our method

can be limited to apply to linear layers pruning, since linear parameters increase exponentially in each layer and hard to mask out. We plan to explore such issue in future study.

## 6 Acknowledgment

This work was supported in part by the NSF under Grant OAC-2320999, CNS-2120279, IIS-2236578, DGE-2336109, CNS-2413009, and SaTC-2439013, CNS-2153358, DoD Center of Excellence in AI and Machine Learning (CoE-AIML) under Contract Number W911NF-20-2-0277, and the Commonwealth Cyber Initiative, the Air Force Research Lab under grant FA8750-19-3-1000, and NSA under grants HQ00342410026 and H98230-21-1-0263.

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

# 7 Appendix

## 7.1 More comparisons

We evaluate our method against DeepReShape Jha & Reagen (2024) by comparing the pruned model accuracy on the ResNet34 architecture under a fixed ReLU budget. This comparison is conducted on both the CIFAR-100 and Tiny ImageNet datasets. Fig.10 and Fig.11 show that our method has around 4.3% more accuracy of pruned(reshaped) model accuracy on same given ReLU budget.

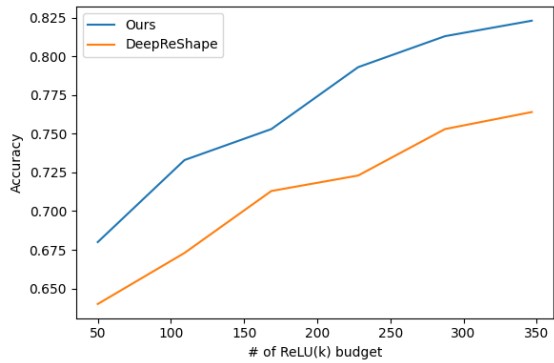

Figure 10: The accuracy of ResNet34 on CIFAR-100, HybReNet for DeepReShape for different given ReLU budgets.

Figure 11: The accuracy of ResNet34 on Tiny-Imagenet, HybReNet for DeepReShape for different given ReLU budgets.

## 7.2 Evidence of Marginal Gain

To provide some evidence for $U(S' \cup i) - U(S') \leq \beta$, where $\beta \ll \alpha$ in Section. 3.1, we sample 100 descendant subset's the marginal utility gain $U(S' \cup i) - U(S')$ under an ancestor node that satisfies $U(S) < 5\%$ for ResNet-9, 18, 34 models in CIFAR 100. The Fig. 13, 15 and 17 show that the marginal utility can be smaller in two order of magnitude of $U(S)$.

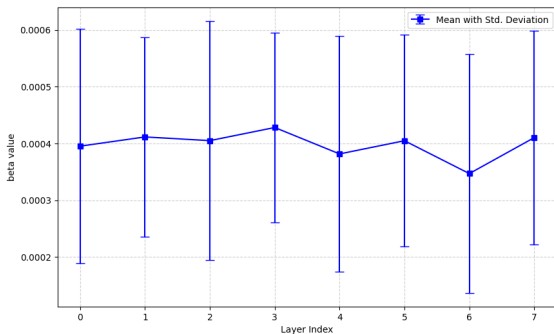

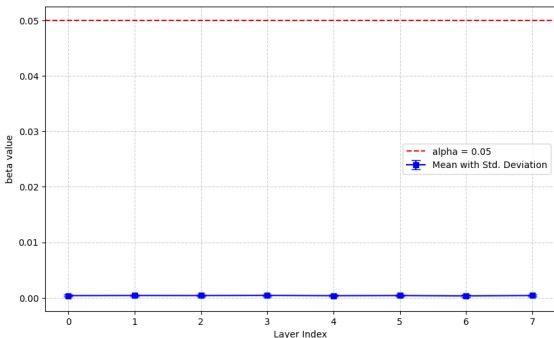

Figure 12: The 100 random samples of $U(S' \cup i) - U(S')$ value distribution of each $i$-th ReLU layers in ResNet-9 model (CIFAR-100).

Figure 13: The zoom-out comparison with each ReLU layer marginal gain and threshold $\alpha$ in ResNet-9 (CIFAR-100).

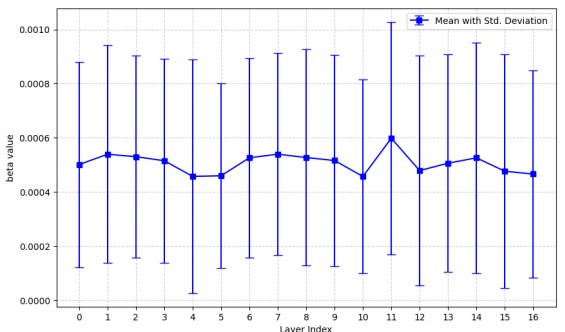

Figure 14: The 100 random samples of $U(S' \cup i) - U(S')$ value distribution of each $i$-th ReLU layers in ResNet-18 model (CIFAR-100).

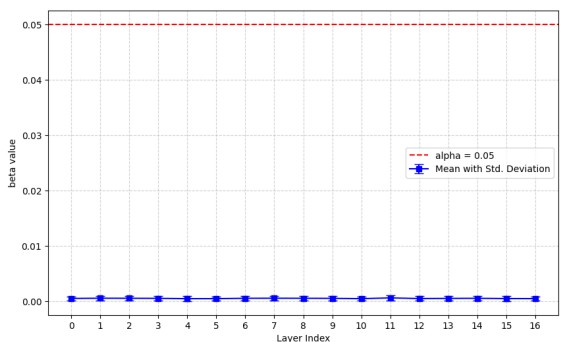

Figure 15: The zoom-out comparison with each ReLU layer marginal gain and threshold $\alpha$ in ResNet-18 (CIFAR-100).

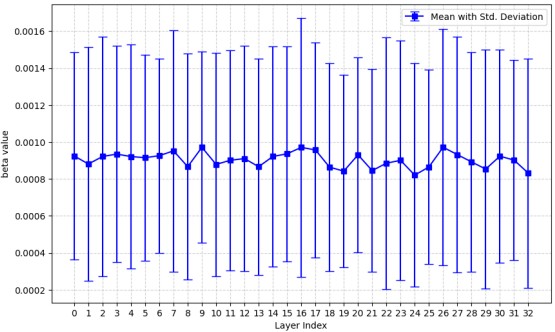

Figure 16: The 100 random samples of $U(S' \cup i) - U(S')$ value distribution of each $i$-th ReLU layers in ResNet-34 model (CIFAR-100).

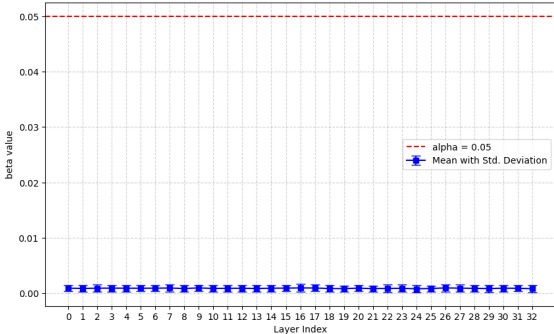

Figure 17: The zoom-out comparison with each ReLU layer marginal gain and threshold $\alpha$ in ResNet-34 (CIFAR-100).

