# OpenReview forum: "PrivShap: A Finer-granularity Network Linearization Method for Private Inference"
_TMLR — Accepted by TMLR_

### Review · Reviewer_DU4C · 2025-06-20

**Summary Of Contributions:**

This paper focuses on communication-efficient private inference of ReLU networks via layer pruning. To this end, the authors propose to calculate the ReLU only for the top-$k$ layers in terms of their Shapley values. To combat the high computational cost of computing the exact Shapley values, the authors propose to approximate them by greedily construct a tree with a network with one layer, say layer $i$, removed as its root. A tree is then constructed by iteratively removing a network's layer, and the branch to any node (any network) with low accuracy is removed. Shapley value for layer $i$ is then calculated using only the set of visible nodes in the tree, which is much smaller than the set of all sub-networks of the root network.

**Audience:**

Yes

**Broader Impact Concerns:**

There is no ethical concerns.

**Claims And Evidence:**

Yes

**Requested Changes:**

In Section 2.3, please refer to the specific section in the cited paper that contains the proofs of Zero/Negative Contribution and Efficiency.

**Strengths And Weaknesses:**

## Strengths

- The method is clearly illustrated in the paper.
- The method is relatively simple, which is nice. And the results of the experiments, specifically Figure 4, are very convincing.

## Weaknesses

I do not find any critical flaw in this paper. But I'm quite new to this line of work so I'll wait to to see other reviewers' comments. Nonetheless, I have a couple of comments:

- In the experiments, can the authors compare the proposed method with the recent private inference methods from Jah and Reagan (2024) and Sarkar, Kundu & Beereil (2024)?
- In Table 4, why do you use Pearson correlation as the performance metric? Aren't distance based metric like $\ell^2$ loss better for measuring the approximation quality?
  Also, the values of $\rho$ is this table are indistinguishable without standard deviations from repeated trials.
- For the proof in page 6, doesn't the conclusion immediately follow from the LHS of Eq. (2) since $\binom{|S|}{|S'|} \leq \binom{|N|-1}{|S'|}$?


References:
Jha, N. K., & Reagen, B. (2024). DeepReShape: Redesigning Neural Networks for Efficient Private Inference. Transactions on Machine Learning Research. Retrieved from https://openreview.net/forum?id=iwCBWULItx
Sarkar, S., Kundu, S., & Beereil, P. A. (2024). RLNet: Robust Linearized Networks for Efficient Private Inference. 2024 IEEE/CVF Conference on Computer Vision and Pattern Recognition Workshops (CVPRW), 244–253. doi:10.1109/CVPRW63382.2024.00029

---

> ### Author Response · Authors · 2025-07-11
>
> - In the experiments, can the authors compare the proposed method with the recent private inference methods from Jah and Reagan (2024) and Sarkar, Kundu & Beereil (2024)?
>
> We thank the reviewer for pointing out these similar works. While the work of Sarkar, Kundu & Beereil (2024) also deals with model optimization, it focuses on a different application. Our goal is to prune models for faster private inference, utilizing a hybrid approach of Homomorphic Encryption (HE) and Multi-Party Computation (MPC). Their work, on the other hand, explores the trade-off between the speed of inference and its vulnerability to backdoor attacks. Given these divergent research objectives, we think it is hard to compare them fairly.
>
> We have a more similar goal to Jah and Reagan (2024), which is to make private inference faster by pruning the model. Their work reshapes the model to balance the efficiency of linear operations (measured in FLOPs) and activations. Their approach builds on a previous assumption (from SNL paper) that shallow activation layers are less critical, using this idea to guide their model reshaping could hinder the process to the optimized reshaped model. We have conducted a comparison and added the results to our appendix 6.1 (see Figure 10, 11). Results show that our method can achieve around 4.3\% accuracy increase with the same ReLU budget level.
>
>
>
>
> - In Table 4, why do you use Pearson correlation as the performance metric? Aren't distance based metric like loss better for measuring the approximation quality? Also, the values of is this table are indistinguishable without standard deviations from repeated trials.
>
> We thank the reviewer for suggestion on $l_2$ norm. The reason we choose the pearson correlation metric is that the approximated shapley values are the indicators for allocating activation budget. The ranking of Shapley values are more important than the absolute magnitute, since the layerwise allocated activation quantity is given based on the normalized approximated shapley value rather than its magnitude. Thus, we use pearson correlation to reflect the trendence of approximated and original shapley values among layers.
>
> We thank the reviewers for the deviation suggestion. We have updated the table with deviation of 5 trials of experiments and updated Table 4 in the new version manuscript.
>
> - For the proof in page 6, doesn't the conclusion immediately follow from the LHS of Eq. (2)?
>
> The reviewer is correct. Yet, we try to make our proof clear as much as possible, so we prefer to provide step-by-step proof to enhance our paper's readability.
>
>
> - In Section 2.3, please refer to the specific section in the cited paper that contains the proofs of Zero/Negative Contribution and Efficiency.
>
> We thank the reviewer for suggestion and update the manuscript in Sec. 2.3 with blue texts.

---

> > ### Comment · Reviewer_DU4C · 2025-07-11
> > **Thanks**
> >
> > Thanks for the response. Almost all of my concerns have been addressed, though I don't quite agree with the following point regarding the proof in Page 6:
> >
> > > Yet, we try to make our proof clear as much as possible, so we prefer to provide step-by-step proof to enhance our paper's readability.
> >
> > My suggestion is that we can finish the proof at the second line like so:
> >
> > $$  \frac{\beta}{\lvert N \rvert} \sum_{0 \leq k \leq \lvert S \rvert} \frac{\binom{\lvert S \rvert}{k}}{\binom{\lvert N \rvert - 1}{k}} \leq  \frac{\beta}{\lvert N \rvert} \sum_{0 \leq k \leq \lvert S \rvert} 1 = \beta\frac{\lvert S \rvert + 1}{N}.$$
> >
> > It should be generally known that $\binom{\lvert S \rvert}{k} \leq \binom{\lvert N \rvert - 1}{k}$ for all $k$. Sometimes longer proof doesn't make it clearer, which I think is the case here.

---

> ### Author Response · Authors · 2025-07-12
>
> We thank the reviewer's further explanation for us better understanding on proof section.
> We agree with the reviewer's suggestion on shorten proof section for clearness and will update accordingly in the next version with other necessary changes from other reviewers' response.

---

> > ### Comment · Reviewer_DU4C · 2025-07-17
> > **Thanks**
> >
> > Thanks! I'm looking forward to the next version.

---

> > > ### Author Response · Authors · 2025-07-18
> > >
> > > We updated the proof section in new version as reviewer suggests and appreciate the valueable review.

---

### Review · Reviewer_R74T · 2025-06-24

**Summary Of Contributions:**

This maniscript addresses the critical challenge of reducing communication overhead in private inference (PI) caused by non-linear operations like ReLU. The authors propose PrivShap , a novel method leveraging Shapley values from cooperative game theory to assess the layer-wise importance of ReLU layers and allocate budgets for pruning. To mitigate the computational complexity of exact Shapley value calculation, they introduce a tree-trimming algorithm for efficient estimation and a gradient-projection pruning strategy to enforce budget constraints. Extensive experiments on diverse models (ResNet, WideResNet, VGG) and datasets (CIFAR, Tiny-ImageNet) demonstrate superior accuracy-efficiency trade-offs compared to existing methods like SENet and SNL.

**Audience:**

Yes

**Claims And Evidence:**

Yes

**Requested Changes:**

Please refer to the Weaknesses.

**Strengths And Weaknesses:**

**Strengths**
- The maniscript uses Shapley values to quantify ReLU layer importance, moving beyond heuristic assumptions,  which provides a theoretically grounded framework for fine-grained budget allocation.

- The maniscript provides bounds on the error introduced by tree trimming, ensuring that the approximation does not significantly perturb Shapley value estimates.


**Weaknesses**
- The method relies on manually setting a trimming threshold of 5% to prune subsets. While experiments show stability within varying values (5, 10, 15%), an adaptive thresholding mechanism could improve robustness across architectures.

- While the tree-trimming algorithm improves efficiency, its applicability to large-scale models remains untested.
- In Table 5, why do the wall-clock time and pruned model accuracy of ResNet-18 and ResNet-34 appear to be identical? Could the authors clarify whether this reflects experimental consistency?

---

> ### Author Response · Authors · 2025-07-11
>
> - The method relies on manually setting a trimming threshold of 5% to prune subsets. While experiments show stability within varying values (5, 10, 15%), an adaptive thresholding mechanism could improve robustness across architectures.
>
> We're grateful for the reviewer's insightful suggestion regarding adaptive thresholding. We concur that implementing an adaptive algorithm is a promising way to enhance the method's robustness, and we've noted this as a key area for future investigation.
>
>
> - While the tree-trimming algorithm improves efficiency, its applicability to large-scale models remains untested.
>
> We thank the reviewer's suggestion. We have added the experiment on larger scale models e.g., ResNet50/101/152 as follows.
> | Model | Methods | Wall-clock time | Pruned model accuracy |  |  |
> |:---:|:---:|:---:|:---:|:---:|:---:|
> |  |  |  | C-100 | C-10 | T-imagenet |
> | ResNet-101 | Ours | **~76 min** | 79.5\% | 96.3\% | 68.3\% |
> |  | MC | $> 375$ min | 75.6\% | 92.8\% | 64.2\% |
> |  | MAB | $> 316$ min | 75.4\% | 91.5\% | 66.1\% |
> | ResNet-152 | Ours | **~124 min** | 77.3\% | 96.9\% | 69.2\% |
> |  | MC | $> 410$ min | 73.1\% | 93.1\% | 62.7\% |
> |  | MAB | > 396  min | 76.1\% | 90.9\% | 67.2\% |
>
>
> - In Table 5, why do the wall-clock time and pruned model accuracy of ResNet-18 and ResNet-34 appear to be identical? Could the authors clarify whether this reflects experimental consistency?
>
> We thank the reviewer for pointing this out. We apologize for these errors and have corrected them in the revised manuscript.

---

> > ### Comment · Reviewer_R74T · 2025-07-13
> > **Comment on the Response**
> >
> > Thank you for your response, which has addressed most of my concerns. I would still encourage the authors to release the code to facilitate reproducibility.

---

### Review · Reviewer_TeZD · 2025-07-02

**Summary Of Contributions:**

Due to the high computational overhead of ReLU activations in private inference, this paper introduces a ReLU pruning method to reduce their cost. The authors propose a tree-trimming algorithm based on Shapley values to estimate the importance of each ReLU layer. Using these importance values, they allocate a computational budget to each layer and apply gradient-projection pruning to selectively remove ReLUs at finer granularities (e.g., pixel and channel levels).

The authors, in their experiments on three datasets and different model architectures, show that their method results in fewer ReLU operations and overall has a better performance compared to the SOTA methods with a similar ReLU budget.

**Audience:**

Yes

**Claims And Evidence:**

Yes

**Requested Changes:**

1. This one is more of a question than a request since I'm not super familiar with HE methods. I wonder if instead of removing ReLU activations, we can replace them with other non-linear activations. Would that have the same computational overhead issue? Or are there other reasons that authors and other papers decided to completely remove ReLU activations?
2. Table 1 is not referenced anywhere and needs to be explained in the main text.
3. The theoretical analysis shows that if $U(S' \cup \{i\}) - U(S') \leq \beta$ for all $S' \subseteq S$, then $\Delta \text{SV}_i \leq \beta \frac{|S| + 1}{|N|}$ and the authors concluded we can prune and nodes closer to the root and avoid the exponential computations with negligible purterbation but it is all based on having  $\beta \ll \alpha$ which is not shown anywhere in the paper why it is true. Can you provide some evidence for that?
4. In Figure 1, the number for one of the dashed arrows is missing.
5. I recommend that the authors provide a bit more explanation at the beginning of Section 3, before introducing Shapley value details, to give a clearer overview of how their method works. Specifically, it would help to clarify that the approach involves two steps: first, using Shapley values and tree-trimming to allocate a ReLU budget across layers, and second, applying gradient-projection pruning to remove activations at a finer granularity. This structure is not immediately clear from the current text, and I had to reread the section several times to fully understand it.
6. Section 3.2 would benefit from additional explanation, detail, and mathematical formulation. For instance, the dimensions d, m, and n in $x_i \in \mathbb{R}^{d \times m \times n}$ are not defined—what do they represent? It’s also unclear from which objective the $l_1$-norm is being removed. Additionally, the gradient descent step is not sufficiently explained. Even if these components are based on prior work, since they are integral to your method, you should provide the necessary details to ensure clarity and self-containment for the reader.
7. In Table 2 and 3, model architectures are different and some of the methods are not comparable. Can you please explain why?
8. In the first paragraph of page 10, you mention your method has more communication savings for ResNet-18 in CIFAR-10, but I do not see any ResNet-18 for your method in the table.

**Strengths And Weaknesses:**

Strength:

The paper introduces a novel method for ReLU pruning and demonstrates its effectiveness through experiments, showing that it outperforms state-of-the-art approaches. The authors also provide a theoretical analysis to support the validity of their tree-trimming algorithm.

Weaknesses:

Mostly smaller weaknesses that I list here:
1. The writing could be improved, particularly in the methods section. For example, Section 3.2 feels somewhat rushed and would benefit from clearer explanations or additional detail. Please check the requested changes section for more details.
2. In the theoretical analysis section, there is a strong statement without any evidence or proof: "We start with a natural observation that $\forall S' \subseteq S$ and $U(S) \leq \alpha$, , then the marginal gain satisfies  $U(S' \cup \{i\}) - U(S') \leq \beta$, where $\beta \ll \alpha$." And the whole proof is based on this statement.
3. In some of the experiments shown in Tables 2 and 3, it is hard to compare different methods since they have different architectures. I wonder why the authors decided to compare their method that way.

---

> ### Author Response · Authors · 2025-07-11
>
> - This one is more of a question than a request since I'm not super familiar with HE methods. I wonder if instead of removing ReLU activations, we can replace them with other non-linear activations. Would that have the same computational overhead issue? Or are there other reasons that authors decided to completely remove ReLU activations?
>
> We thank the reviewer for this interesting question. Homomorphic Encryption schemes primarily support simple arithmetic operations like addition and multiplication on encrypted integers. Consequently, complex non-linear activation functions such as ReLU and GeLU cannot be implemented directly. They must be approximated by polynomials. ReLU function is the most "crypto-friendly" due to its simple structure, which allows for a more efficient polynomial approximation compared to other functions. However, it still incurs large amount of communication burden. Switching to a more complex function like GeLU would require a higher-degree polynomial for approximation, which would only worsen the performance bottleneck. Thus, our paper, as long as other similar works, try to remove less important pixel-wise ReLU to reduce the latency.
>
>
>
> - Table 1 is not referenced anywhere and needs to be explained in the main text.
>
> We thank the review and add the context to refer Table 1 in the updated maunscript.
>
> - The theoretical analysis shows that if  for all , then and the authors concluded we can prune and nodes closer to the root and avoid the exponential computations with negligible purterbation but it is all based on having which is not shown anywhere in the paper why it is true. Can you provide some evidence for that?
>
>
> We thank the reviewer for the careful review. We implictly mentioned the such observation in Sec. 3.1 in "This also makes such low accuracy model irreversible to high accuracy model by just adding one important activation layer." To demonstrate some evidence to this, we conduct additional experiments on sampling 100 descendant subset's the marginal utility $U(S' \cup i) - U(S')$ under an ancestor layer node that its $U(S)<5\%$ for ResNet-9,18,34 in CIFAR-100 dataset. The result figures are updated in appendix 6.2 and results show that the marginal utility is usually smaller in two order of magnitude of $U(S)$. Please check the details in Fig 13, 15, 17.
>
>
>
> - In Figure 1, the number for one of the dashed arrows is missing.
>
> Thank you for the review. The omission of number for subset {1,5} is intentional, as it was evaluated in a preceding node (indicated on the left of the same row with an accuracy of 6.1%). To avoid redundant computations, the algorithm bypasses nodes that have been previously explored.
>
>
> - I recommend that the authors provide a bit more explanation at the beginning of Section 3, before introducing Shapley value details, to give a clearer overview of how their method works. Specifically, it would help to clarify that the approach involves two steps: first, using Shapley values and tree-trimming to allocate a ReLU budget across layers, and second, applying gradient-projection pruning to remove activations at a finer granularity. This structure is not immediately clear from the current text, and I had to reread the section several times to fully understand it.
>
>
> We thank the reviewer for the suggestion and updated with the corresponding blue text at the beginning of Section 3.
>
>
> - Section 3.2 would benefit from additional explanation, detail, and mathematical formulation. For instance, the dimensions d, m, and n are not defined—what do they represent? It’s also unclear from which objective the l1-norm is being removed. Additionally, the gradient descent step is not sufficiently explained. Even if these components are based on prior work, since they are integral to your method, you should provide the necessary details to ensure clarity and self-containment for the reader.
>
>
> We thank the suggestions from reviewers and update the corresponding parts in new verison with blue texts.
>
>
>
> - In Table 2 and 3, model architectures are different and some of the methods are not comparable. Can you please explain why?
>
> We thank the reviewer for their valuable feedback. We acknowledge the reviewer's point about direct, head-to-head comparisons. A simple side-by-side comparison is challenging because key continuous metrics such as the final number of ReLUs, ReLU budgets, inference latency, and model accuracy are difficult to be aligned across different studies to create a perfectly controlled, apples-to-apples comparison. Instead, we choose similiar scale of parameter and model for comparison, to both save the experiment burden and demonstrate performance. In addition, we anchor the performance of our method and all baselines to the "All-ReLU" model for latency comparison. Our goal is to comprehensively demonstrate our method's effectiveness on the network linearization problem across various baseline methods, model architectures, datasets, and ReLU budgets.

---

> > ### Author Response · Authors · 2025-07-11
> > **Continuing the response**
> >
> > - In the first paragraph of page 10, you mention your method has more communication savings for ResNet-18 in CIFAR-10, but I do not see any ResNet-18 for your method in the table.
> >
> > We apologize for the typo on the text. It should be ResNet-101 and is corrected in updated version.

---

> > > ### Comment · Reviewer_TeZD · 2025-07-26
> > >
> > > I cannot see the correction.

---

> > > > ### Author Response · Authors · 2025-07-26
> > > >
> > > > Thanks for your response, Could you specify which correction you are referring to that cannot see?
> > > > The corrections are in the updated pdf version. You may re-open the new pdf manuscripts for the change.
> > > > The blue text in the beginning of Section 3 and in Section 3.2 are the changes.

---

> > > > > ### Comment · Reviewer_TeZD · 2025-07-26
> > > > >
> > > > > Sorry for not clarifying. I meant this typo:
> > > > > "Our method also demonstrates 1.3×ReLU budgets compared to the SOTA and up to 21.7× communication savings for ResNet-18 in CIFAR-10, as shown in Table 3", last paragraph of page 9. I think you meant to say ResNet-101 here?

---

> > > > > > ### Author Response · Authors · 2025-07-26
> > > > > >
> > > > > > Sorry for the incomplete updates in the new maunscript.
> > > > > > Yes, we meant to say ResNet-101 here and will update this with uploading a new pdf very soon.
> > > > > > Please feel free to ask any further questions or corrections if have and we will update the maunscript to save multiple notifications.

---

> > ### Comment · Reviewer_TeZD · 2025-07-26
> >
> > Thank you for addressing my comments. I also agree with reviewer "R74T" that releasing your code would be super useful.

---

> > > ### Author Response · Authors · 2025-07-26
> > >
> > > Thanks all reviewers for careful and valueable reviews.
> > > We are working on organizing the code and will publish the implementation soon for further study.

---

### Decision · Action_Editor_VzPr · 2025-08-04

**Recommendation:** Accept as is

**Audience:**

Yes

**Audience Explanation:**

The paper addresses a well-studied problem of private inference that is definitely interesting to the TMLR audience. Reviewers all find the method interesting and its result convincing.

**Claims And Evidence:**

Yes

**Claims Explanation:**

The authors propose a new method based on Shapley value for pruning activations in ReLU layers. Empirical result shows that PrivShap outperforms other private inference methods in terms of accuracy and communication cost. Reviewers generally appreciate the method's novelty and theoretical soundness. Although it is only demonstrated on small-scale models and datasets, this does not affect the accuracy of the paper's claims.